# Use of Mass Spectrometry to Determine the Diversity of Toxins Produced by *Gambierdiscus* and *Fukuyoa* Species from Balearic Islands and Crete (Mediterranean Sea) and the Canary Islands (Northeast Atlantic)

**DOI:** 10.3390/toxins12050305

**Published:** 2020-05-07

**Authors:** Pablo Estevez, Manoëlla Sibat, José Manuel Leão-Martins, Angels Tudó, Maria Rambla-Alegre, Katerina Aligizaki, Jorge Diogène, Ana Gago-Martinez, Philipp Hess

**Affiliations:** 1Biomedical Research Center (CINBIO), Department of Analytical and Food Chemistry, Campus Universitario de Vigo, University of Vigo, 36310 Vigo, Spain; paestevez@uvigo.es (P.E.); leao@uvigo.es (J.M.L.-M.); 2Laboratoire Phycotoxines, Ifremer, Rue de l’Île d’Yeu 44311 Nantes, France; manoella.sibat@ifremer.fr; 3Marine and Continental Waters programme, Ctra. Poble Nou, km. 5.5, IRTA, Sant Carles de la Ràpita, 43540 Tarragona, Spain; angels.tudo@irta.cat (A.T.); maria.rambla@irta.cat (M.R.-A.); 4Laboratory Unit on Harmful Marine Microalgae, Biology Department, Aristotle University of Thessaloniki, 54124 Thessaloniki, Greece; aligiza@auth.gr

**Keywords:** maitotoxins, ciguatoxins, *Gambierdiscus*, *Fukuyoa*, LC-MS/MS, HRMS, QToF, ciguatera poisoning

## Abstract

Over the last decade, knowledge has significantly increased on the taxonomic identity and distribution of dinoflagellates of the genera *Gambierdiscus* and *Fukuyoa*. Additionally, a number of hitherto unknown bioactive metabolites have been described, while the role of these compounds in ciguatera poisoning (CP) remains to be clarified. Ciguatoxins and maitotoxins are very toxic compounds produced by these dinoflagellates and have been described since the 1980s. Ciguatoxins are generally described as the main contributors to this food intoxication. Recent reports of CP in temperate waters of the Canary Islands (Spain) and the Madeira archipelago (Portugal) triggered the need for isolation and cultivation of dinoflagellates from these areas, and their taxonomic and toxicological characterization. Maitotoxins, and specifically maitotoxin-4, has been described as one of the most toxic compounds produced by these dinoflagellates (e.g., *G. excentricus*) in the Canary Islands. Thus, characterization of toxin profiles of *Gambierdiscus* species from adjacent regions appears critical. The combination of liquid chromatography coupled to either low- or high-resolution mass spectrometry allowed for characterization of several strains of *Gambierdiscus* and *Fukuyoa* from the Mediterranean Sea and the Canary Islands. Maitotoxin-3, two analogues tentatively identified as gambieric acid C and D, a putative gambierone analogue and a putative gambieroxide were detected in all *G. australes* strains from Menorca and Mallorca (Balearic Islands, Spain) while only maitotoxin-3 was present in an *F. paulensis* strain of the same region. An unidentified *Gambierdiscus* species (*Gambierdiscus* sp.2) from Crete (Greece) showed a different toxin profile, detecting both maitotoxin-3 and gambierone, while the availability of a *G. excentricus* strain from the Canary Islands (Spain) confirmed the presence of maitotoxin-4 in this species. Overall, this study shows that toxin profiles not only appear to be species-specific but probably also specific to larger geographic regions.

## 1. Introduction

Ciguatera Poisoning (CP) is described as a food intoxication endemic in tropical and subtropical areas of the world. The poisoning is caused by the consumption of fish or shellfish that accumulate toxic compounds produced by benthic dinoflagellates of the genus *Gambierdiscus* and *Fukuyoa* [1]. The two main toxin groups produced by these dinoflagellates are ciguatoxins (CTXs) and maitotoxins (MTXs) [2].

CTXs are considered the main toxins responsible for CP as their lipophilic character allows for intestinal absorption and accumulation. They are cyclic polyether compounds of around 1100 Da being classified as Pacific (P-CTXs), Caribbean (C-CTXs) and Indian (I-CTXs) ciguatoxins. Different CTX analogues from these groups have been detected in fish tissue associated to a CP case, while only a few P-CTXs were detected in *Gambierdiscus* extracts from the Pacific Ocean [3,4,5,6,7].

MTXs are water soluble cyclic polyethers containing one or two sulfate ester groups; these groups are responsible for the intermediate polarity of MTXs, and their low intestinal absorption casts doubts as to their involvement in CP [8]. Six MTX analogues have currently been identified: maitotoxin-1 (MTX1), maitotoxin-2 (MTX2), maitotoxin-3 (MTX3), maitotoxin-4 (MTX4), desulfo-MTX1 and didehydro-demethyl-desulfo-MTX1, all of which were isolated from different strains of *Gambierdiscus* [9,10,11,12].

MTX1 is the most toxic marine compound and the largest natural non-biopolymer toxin consisting on a ladder-shaped cyclic polyether compound containing two sulfate groups [9]. MTX2 was isolated from an Australian *Gambierdiscus* strain from Queensland [10]. Its structure has not yet been elucidated and it showed a lower potency than MTX1 by intraperitoneal injection in mice. MTX3 (44-methylgambierone) was first characterized by [10], but it was not until recently that its structure had been elucidated by its isolation from *G. belizeanus* and *G. australes*, being identified as a gambierone homologue [13,14,15]. MTX3, which is about one third the molecular weight of MTX1, showed a biological activity similar to CTX3C but with much lower potency, indicating that despite being grouped in the MTX group, this compound exhibits CTX-like activity rather than MTX-like activity [13]. MTX4 was recently isolated from *G. excentricus* extracts from the Canary Islands (Spain) [11]. Its structure is not yet elucidated but it has ion clusters and molecular mass in a similar range as MTX1, as well as sulfate esters; it was reported to exhibit a similar toxic effect of MTX1 in neuroblastoma cells detecting a high cytotoxicity and Ca^2+^ influx [11]. Other large but mono-sulfated MTXs were recently elucidated by their isolation from *Gambierdiscus* spp. from the Caribbean Sea, i.e., desulfo-MTX1 from *G. belizeanus* and didehydro-demethyl-desulfo-MTX1 from *G. ribotype-2*, showing the wide variety of MTXs that seem to be produced by these dinoflagellates [12] (Table 1). Furthermore, *Gambierdiscus* and *Fukuyoa* have shown to produce non-structurally related cyclic polyether compounds such as gambierol, gambieric acids, gambieroxide and gambierone [15,16,17,18].

Most of these compounds can be classified depending on their mechanism of action into two groups: (i) MTX-like compounds, associated with a massive Ca^2+^ influx causing a rapid cell death, including MTX1, MTX2 and MTX4; and (ii) CTX-like compounds, which create a disequilibrium in the voltage-gated sodium channels (VGSCs), including all CTXs, and with much lower potency, MTX3 and gambierone [13].

A wide variety of *Gambierdiscus* species are present in regions considered endemic for CP (the Pacific Ocean or the Caribbean Sea) [19]. These dinoflagellates have also been detected in CP-emerging regions such as the Canary Islands (Spain) and Madeira archipelago (Portugal), as well as in the Mediterranean Sea, from which no CP cases have been reported until now [20]. *Gambierdiscus excentricus* was the first *Gambierdiscus* species detected in the Canary Islands [21], but its detection led to further research that concluded with the detection of a large number of *Gambierdiscus* species (*G. australes, G. caribaeus, G. carolinianus, G. excentricus, G. silvae* and *Gambierdiscus* ribotype3) suggesting that the Canary Islands would be a “hot spot” for these dinoflagellates [22]. On the other hand, the only available data from the Selvagem Islands (Madeira, Portugal) is the detection of numerous strains of *G. australes*, which test positive in the Neuro2a cell-based toxicity assay (N2a), suggesting they contain CTX- and MTX-like compounds [23]. In the Mediterranean Sea, the presence of *Gambierdiscus* was first reported in 2003 (*Gambierdiscus* sp.) in Crete (Greece), while to our knowledge only one study in 2016 attempted to characterize a strain of *F. paulensis* from Formentera (Balearic Islands, Spain) by LC-MS, however, the trace levels of toxic compounds present did not allow for conclusive results about the compounds produced [24,25].

As part of the ongoing EuroCigua project [26], *G. australes* and *F. paulensis* were isolated from Menorca and Mallorca (Balearic Islands, Spain), *Gambierdiscus* sp.2 from Crete (Greece), as well as *G. excentricus* from La Gomera (Canary Islands, Spain), showing CTX and MTX-activity. The aim of the present study was to describe the diversity of toxins produced by a selection of these strains of *Gambierdiscus* and *Fukuyoa* species using liquid chromatography coupled to low- and high-resolution mass spectrometry (LC-MS/MS and LC-HRMS). While LC-MS/MS was used for a rapid screening of the toxic compounds potentially present, allowing for the estimation of their concentrations, the purpose of LC-HRMS was the confirmation and structural characterization of the toxic compounds previously identified by LC-MS/MS.

## 2. Results

### 2.1. LC-MS/MS Analysis

In an attempt to separate MTXs and gambierones from CTXs, liquid/liquid partitioning of the crude extracts was carried out between MeOH:H_2_O (3:2) and dichloromethane. The methanol-soluble fraction (MSF) and the dichloromethane-soluble fraction (DSF) of each crude extract of dinoflagellate cell pellets were analyzed by LC-MS/MS monitoring MTXs, as well as gambierone (Table 2). A Multiple Reaction Monitoring (MRM) method in negative ionization mode screened for six MTXs (MTX1, MTX2, MTX3, MTX4, desulfo-MTX1 and didehydro-demethyl-desulfo-MTX1), as well as gambierone. The ESI^−^ MRM method was based on monitoring a qualitative transition from the (pseudo-)molecular ion to the hydrogenated sulfate anion *m/z* 96.9 [HOSO_3_]^−^ at high collision energy, which is typical of these compounds, and for quantitation based on monitoring the pseudo-molecular ion itself [M−2H]^2−^/[M−2H]^2−^ for MTX1, MTX4, desulfo-MTX1 and didehydro-demethyl-desulfo-MTX1. This was due to the single-charged ion of these compounds being outside the mass range of the mass spectrometer (50–2800 Da), whereas [M-H]^−^/ [M-H]^−^ was selected for MTX3 and gambierone. Due to the absence of MTX2 MS/MS fragmentation data, the double-charged [M−2H]^2−^/[HOSO_3_]^−^ and triple-charged [M−3H]^3−^/[HOSO_3_]^−^ molecular anions to the sulfate anion *m/z* 96.9 [HOSO_3_]^−^ were selected as MRM transitions for this compound, assuming similar MS/MS fragmentation behavior of MTX1 and MTX2 [27].

The identification of MTX1, MTX3, gambierone and MTX4 was carried out by comparing retention times as well as ion ratios with the reference materials available (Appendix A). On the other hand, the lack of structural information and reference material of MTX2 limited its identification. Therefore, the absence of this compound was reported as not detected above the limit of detection of any ion transition monitored. The lack of reference material of desulfo-MTX1 and didehydro-demethyl-desulfo-MTX1 also limited its identification, with the approach followed being the same as that described for MTX2. However, the accurate mass of these compounds was available, only hampering its identification in LC-HRMS by the uncertainty of the limit of detection of our method for these compounds.

44-methyl-gambierone (MTX3) was identified in all samples of *G. australes* from Menorca and Mallorca (Balearic Islands, Spain), detecting both quantitative *m/z* 1037.6 [M-H]^−^/ *m/z* 1037.6 [M-H]^−^ and qualitative *m/z* 1037.6 [M-H]^−^/ *m/z* 96.9 [HOSO_3_]^−^ MRM ion transitions in negative ionization mode. MTX3 concentrations in *G. australes* strains ranged from 344 to 1661 pg MTX1 equivalent/cell (eq./cell) taking into account the total amount detected in the sum of MSF + DSF. *Gambierdiscus* sp.2 from Crete (Greece) contained significantly less MTX3 (4.3 pg MTX1 eq./cell). The recovery of MTX3 in the MSF ranged between 66.7% and 84.1% for *G. australes* for the Balearic and Crete strains (Table 2). MTX3 was also present in one *F. paulensis* from Menorca (Balearic Islands, Spain) at a concentration of 10.5 pg MTX1 eq./cell. In this case, however, the recovery of MTX3 in the MSF was only 17.7% (Table 2). This difference compared to the recovery observed in the MTX3 detected in *G. australes* must be explored; at this stage, we can only presume a matrix effect during the partitioning step. Gambierone was only identified in the one *Gambierdiscus* strain from Crete, albeit at a significant concentration of 776 pg MTX1 eq./cell for the sum of MSF + DSF. The recovery of gambierone in the MSF was 92.2% (Table 2).

A putative gambierone analogue, with an earlier retention time than gambierone on the C_18_ column, was detected in all *G. australes* from Menorca and Mallorca (Balearic Islands, Spain) at concentrations ranging from 148.6 to 523.4 pg MTX1 eq./cell and an average recovery of the putative gambierone analogue in the MSF of 99.7% (Table 2). Negative ionization mode detected both quantitative *m/z* 1023.5 [M-H]^−^/ *m/z* 1023.5 [M-H]^−^ and qualitative ion transitions *m/z* 1023.5 [M-H]^−^/m/z 96.9 [HOSO_3_]^−^ with the same ion ratios as gambierone. This compound was not present in the other three strains of *Gambierdiscus*.

Further confirmation of MTX3, gambierone and the putative gambierone analogue was carried out in positive ionization mode and not only in MRM, but also in full scan and enhanced product ion modes, confirming the presence of MTX3 and gambierone due to the detection of their common fragment *m/z* 803 as well as their specific fragments *m/z* 233 and *m/z* 219, respectively. This was already reported by Boente-Juncal et al. [13], and a common fragment with *m/z* 109 was assigned to the fragmentation of the side chain in the last ring (Appendix A). The putative gambierone analogue did not show any common or specific gambierone fragments but showed a fragmentation pattern similar to these compounds, including water losses and sulfate loss followed by water losses (Appendix A).

MTX4 was confirmed in a strain of *G. excentricus* from La Gomera (Canary Islands, Spain). Retention time as well as MRM ion ratios transitions *m/z* 1646.2 [M−2H]^2−^/*m/z* 1646.2 [M−2H]^2−^ and *m/z* 1646.2 [M−2H]^2−^/*m/z* 96.9 [HOSO_3_]^−^ were consistent with those obtained in the MTX4 reference material. The recovery of MTX4 from the DSF to the MSF was 99.3% (See Table 2).

MTX1, MTX2, desulfo-MTX1 and didehydro-demethyl-desulfo-MTX1 were not detected in any *Gambierdiscus* or *Fukuyoa* samples from the Mediterranean Sea and the *G. excentricus* from Canary Islands (Spain), neither in the MSF nor in the DSF.

### 2.2. LC-HRMS/MS Analysis

The presence of the compounds previously identified and quantified by LC-MS/MS was confirmed by LC-HRMS. Mass spectral detection was performed in full scan and targeted MS/MS mode in negative (ESI^−^) and positive (ESI^+^) ionization mode. Both DSF and MSF of dinoflagellate extracts were analyzed in ESI^−^ and ESI^+^ full scan mode, and screening with an in-house database allowed for identification of several compounds based on their characteristics reported in the literature.

MTX3 was confirmed in all *G. australes* in both ESI^−^ and ESI^+^ full scan mode at 7.6 min. Negative ionization mode showed the detection of a single ion corresponding to the deprotonated molecule [M-H]^−^ with ∆ppm < 1 ppm in all samples of this species (Figure 1A1, Appendix A). At the same retention time, positive ionization mode showed a prominent ion corresponding to the molecular ion [M+H]^+^ and also pseudo-molecular ions [M+H-H_2_O]^+^, [M+NH_4_]^+^, [M+Na]^+^ with ∆ppm < 3.7 ppm in all ions of these samples (Figure 1A) (See Appendix A).

The molecular ion [M-H]^−^ of MTX3 was also detected in *F. paulensis* and *Gambierdiscus* sp.2 in ESI^−^ ionization mode at 7.6 min with a ∆ppm < 1.9 ppm, whereas the lower sensitivity of the ESI^+^ ionization mode for this compound only allowed the detection of the molecular ion [M+H]^+^ with ∆ppm < 2.2 ppm and traces of the first water loss [M+H-H_2_O]^+^ (∆ppm < 8 ppm) (See Appendix A).

Gambierone was identified at 7.4 min in the *Gambierdiscus* sp.2 from Crete (Greece) showing a similar ion pattern as the one detected for MTX3 in both ESI^−^ and ESI^+^ full scan mode. Negative ionization mode showed the detection of a single ion matching the deprotonated molecule [M-H]^−^
*m/z* 1023.4630 (∆ppm = +0.1 ppm) (Figure 1B1, Appendix A). Positive ionization mode showed, at the same retention time, a prominent ion corresponding to the molecular ion [M+H]^+^
*m/z* 1025.4810 (∆ppm = +3.5 ppm) and also the pseudo-molecular ions [M+H-H_2_O]^+^
*m/z* 1007.4687 (Δppm = +1.8), [M+NH_4_]^+^
*m/z* 1042.5060 (Δppm = +1.9) and [M+Na]^+^
*m/z* 1047.4580 (Δppm = −1.3) (Figure 1B, Appendix A).

MTX3 and gambierone were further confirmed in targeted MS/MS in both ESI^−^ and ESI^+^ mode. ESI^−^-targeted MS/MS selected the deprotonated molecule [M-H]^−^ applying 30 eV, 50 eV and 70 eV, detecting the hydrogen-sulfate fragment [HOSO_3_]^−^ in both compounds (Figure 2A and Figure 2B). An ion representing a water loss [M-H-H_2_O]^−^ was also detected in low abundance, as well as the fragments specific for MTX3, *m/z* 977.4535 (∆ppm = −4.0 ppm) and *m/z* 959.4428 (∆ppm = −4.2 ppm), and those specific for gambierone, *m/z* 963.4392 (∆ppm = −2.6 ppm) and *m/z* 945.4302 (∆ppm = −1.0 ppm). Fragmentation pathways are also proposed on the basis of the observed fragments, including the two common fragments with theoretical *m/z* 899.3741 and *m/z* 839.3529, as well as water losses common to both MTX3 and gambierone (Figure 2A1 and Figure 2B1) (Table 3).

Positive mode ESI^+^-targeted MS/MS of MTX3 and gambierone molecular ion [M+H]^+^, *m/z* 1039.4931 and *m/z* 1025.4774, respectively, at an average CE of 20, 40 and 60 eV allowed for the unambiguously confirmation of both compounds. MTX3-targeted MS/MS spectra showed a series of: 1) water losses and sulfite loss plus water losses (Δppm < 4 ppm), 2) the ion with theoretical *m/z* 803.4212 also common to gambierone, and 3) the ions corresponding to those already reported in LC-MS/MS at *m/z* 233.1553 (Δppm = −1.3 ppm) and *m/z* 109.0645 (Δppm = −2.8 ppm), assigned to the fragmentation of MTX3 I-ring (Figure 3A) (Table 3).

The same fragmentation pattern was observed for gambierone with the detection of water losses and sulfite loss plus water losses (Δppm < 5 ppm), the ion with *m/z* 803.4173 (Δppm = −4.9 ppm), and the gambierone specific ions *m/z* 219.1375 (Δppm = −2.1 ppm) and *m/z* 109.0644 (Δppm = −3.6 ppm) (Figure 3B) (Table 3).

Due to the lack of the molecular formula of MTX4, its confirmation in the *G. excentricus* from the Canary Islands (Spain) could not be carried out by screening with the database. However, the availability of MTX4 reference material allowed for confirmation of this compound by comparing retention time (7.5 min), as well as further ESI^−^-targeted MS/MS by selecting the [M-2H]^2−^ and detection of the two fragment ions [HOSO_3_]^−^ and [SO_3_]^−^ (Figure 4).

The putative gambierone analogue detected by LC-MS/MS was identified by LC-HRMS ESI^+^ full scan analysis using the algorithm Find-By-Molecular-Feature (FBF) to detect, at a retention time of 6.0 min, a prominent ion with *m/z* 1042.4911 assigned to [M+NH_4_]^+^ as well as [M+H]^+^
*m/z* 1025.4632, [M+Na]^+^
*m/z* 1047.4427 and [M+K]^+^
*m/z* 1063.4108 (Appendix A). The mass differences (Δppm) for these ions were higher than 13, indicating that this compound is probably not a gambierone isomer (Appendix A, Appendix A). It was also observed that under the LC-HRMS conditions the ion pattern of this compound is different to both gambierone and the putative gambierone analogue in low- and high-resolution MS, where the prominent ion is the protonated molecule [M+H]^+^. ESI^−^-targeted MS/MS of the [M-H]^−^ of the putative gambierone revealed a hydrogenated sulfate anion loss typical of MTX-like compounds, whereas no specific or common fragments to gambierone or MTX3 were detected between *m/z* 770 and *m/z* 1010 (Appendix A). Positive ionization mode targeted MS/MS of the [M+H]^+^ also revealed a similar fragmentation pattern as gambierone and MTX3 with sulfate loss plus water losses, however, large mass differences (Δppm > 20) were observed compared to gambierone and no fragments common to gambierone were detected (Appendix A and Appendix A).

The screening of raw data with the database of the ESI^−^ and ESI^+^ full scan data also detected a putative gambieroxide at a retention time of 5.3 min in the MSF of all *G. australes* strains (Appendix A). Negative ESI full scan mode allowed for detection of the deprotonated molecule [M-H]^−^ of the putative gambieroxide with Δppm < 3 ppm (Appendix A). Positive ESI full scan acquisition showed a prominent ion at *m/z* 1212.6019 corresponding to [M+NH_4_]^+^ with Δppm = +3.0 ppm, and [M+Na]^+^ was also detected at *m/z* 1217.5553 but with lower intensity and a Δppm = +1.2 ppm (Appendix A). Negative ESI-targeted MS/MS of [M-H]^−^
*m/z* 1193.5572 of the putative gambieroxide only revealed a fragment at *m/z* 453.1964 which was not identified in the molecule [18], whereas ESI^+^-targeted MS/MS of [M+NH_4_]^+^
*m/z* 1212.5983 showed a prominent fragment at *m/z* 1159.5493 corresponding to [M+H-2H_2_O]^+^ (−1.1 ppm) and followed by six water loss molecules. The lack of authentic gambieroxide MS/MS data or standard limited the confirmation of this compound (Appendix A).

As above mentioned, the DSF was also analyzed and compared with the corresponding compounds reported in the literature. Two compounds matching gambieric acid C and D were identified in both the ESI^+^ and ESI^−^ MS full scan in *G. australes*. These compounds partially coeluted on the C_18_-column with retention times of 8.895 min for the compound tentatively identified as gambieric acid C and 8.928 min for the compound tentatively identified as gambieric acid D.

For both putative gambieric acid C and D, the deprotonated molecular cluster [M-H]^−^ was detected with ∆ppm < 2.5 ppm in all samples of *G. australes* (Figure 5A1 and Figure 5B1) (See Appendix A). Positive ESI full scan mode showed at the same retention time three prominent ions corresponding to the molecular ion [M+H]^+^ and also pseudo-molecular ions [M+NH_4_]^+^, [M+Na]^+^ with ∆ppm < 5 ppm in all ions of these samples (Figure 5A and Figure 5B, Appendix A).

Positive ESI-targeted MS/MS selecting gambieric acid C and D [M+H]^+^
*m/z* 1185.6932 and *m/z* 1199.7088 respectively, showing a common fragment with *m/z* 135.1174 assigned to the fragmentation of the side chain containing an ester group (Figure 6). Putative gambieric acid C spectra showed two prominent ions *m/z* 1039.6331 (Δppm = −2.0 ppm) and *m/z* 943.5394 (Δppm = −2.0 ppm), both followed by four water losses (Figure 6A). Putative gambieric acid D spectra showed a similar fragmentation pattern with the detection of *m/z* 1053.6477 (Δppm = −3.0 ppm) and *m/z* 957.5542 (Δppm = −2.9 ppm) both followed by water losses, with the most intense being the first water loss of both ions (Figure 6B).

ESI^−^-targeted MS/MS was also carried out by selecting the deprotonated molecular ions of putative gambieric acid C and D [M-H]^−^, *m/z* 1183.6786 and *m/z* 1197.6943, respectively. Despite the difficulty in fragmentation of the molecules in this ionization mode and the low abundance of the fragments, both compounds showed a similar fragmentation pattern, detecting a single water loss molecule [M-H-H_2_O]^−^
*m/z* 1165.6633 (Δppm = −4.0 ppm) and *m/z* 1179.6758 (Δppm = −6.7 ppm), respectively, a fragment corresponding to cleavage on the alfa carbonyl group of the ester *m/z* 1055.6275 (Δppm = −3.6 ppm) and *m/z* 1069.6389 (Δppm = −7.5 ppm) and a common fragment with *m/z* 112.9852 (Appendix A).

## 3. Discussion

The present study identified and quantified compounds by LC-MS/MS and further confirmed via LC-HRMS the presence of maitotoxins as well as gambierone and other cyclic polyether compounds in dinoflagellates of the genus *Gambierdiscus* and *Fukuyoa* from the Mediterranean Sea and one *G. excentricus* from the Eastern Atlantic (Canary Islands, Spain).

LC-MS/MS negative ionization in MRM mode allowed for the sensitive quantitation of various MTXs. The two mono-sulfated MTXs recently isolated from *Gambierdiscus* of the Caribbean Sea, desulfo-MTX1 and didehydro-demethyl-desulfo-MTX1, were also monitored, assuming that due to their similarities to MTX1 they would have a similar fragmentation pattern monitoring [M−2H]^2−^/[M−2H]^2−^ as a quantitative transition and [M−2H]^2−^/[HOSO_3_]^−^ as qualitative transition. Quantitation of the different MTXs and gambierone was carried out using MTX1, the only commercially available maitotoxin. This quantitation may be most adequate for MTX4 which is a compound chemically similar to MTX1, not only in its molecular weight, but also in its fragmentation pattern. MTX3 (=44-methyl-gambierone) and gambierone, which are one-third the molecular weight of MTX1 and MTX4, were also quantified using this approach as they also contain a sulfate group and with results comparable to a commercially available analogue. This approach is traceable but results in somewhat increased uncertainty on any estimated concentration of these compounds.

Among the dinoflagellate species analyzed in this study, *G. australes* appears to be the species that produces the widest variety of hitherto reported *Gambierdiscus* metabolites: MTX3, putative gambieric acids C and D, a putative gambierone analogue and a putative gambieroxide. All *G. australes* strains analyzed were from the same geographical region, Mallorca and Menorca, which are located in the western Mediterranean (Balearic Islands, Spain). The detection of MTX3 in this species is in agreement with the strains of *G. australes* recently isolated from Raoul Island (New Zealand) [14]. No MTX1 was detected in any strain of *G. australes* from Balearic Islands while, in contrast, this compound was detected by LC-MS/MS with MTX3 in strains of the same species from Kochi (Japan) as well as Macaulay Island (New Zealand) [11,28].

The recovery of MTX3 from the DSF was 75% which indicates that the liquid/liquid partition using aqueous MeOH and dichloromethane must be further optimized to quantitatively separate MTX3 from CTXs. This partition behavior also shows that MTX3 has a somewhat lipophilic character which may result in MTX3 being absorbed and accumulated as CTXs in fish and the human digestive system. A recent study showed the accumulation of MTX1 in carnivorous fish tissues (liver and muscles of snappers) after exposure to *G. australes* [29]. While MTX3 clearly has much lower CTX-like activity compared to CTX3C [13], MTX3 was detected in much higher concentrations, estimated up to 1662 pg MTX1-equiv. cell^−1^, suggesting that MTX3 should be monitored in fish muscle in order to explore its possible role in CP.

The absence of MTX1 and the detection of a putative gambierone analogue in all *G. australes* pointed out another possible difference in the toxin profiles of *G. australes* from the Mediterranean Sea and the Pacific Ocean. The use of LC-HRMS showed the ability to conclude that this compound, which initially seemed to be a gambierone isomer by full scan and MRM mode in LC-MS/MS, is not a gambierone isomer due to its high ∆ppm compared to authentic gambierone. Toxicity of purified fractions containing this compound should be evaluated to check its possible interest for the overall assessment of the toxicity of *G. australes*.

Putative gambieric acids C and D were detected in the DSF of all strains of *G. australes*, but not in any other species. Again, this suggests a strong lipophilic character and the potential to accumulate in fish flesh and potentially pass through the human intestinal barrier. Despite the lack of reference material for gambieric acids C and D to compare retention time, the HRMS ESI^−^ and ESI^+^ full scan and targeted MS/MS analysis showed confident data to conclude that the compounds detected should match these potent antifungal compounds, initially isolated from *G. toxicus* from French Polynesia in 1992 [17,30], i.e., prior to this species being separated into several other species. Gambieric acid D was recently detected in the tissue of a shark involved in a CP in the Indian Ocean, suggesting the stability of some of these compounds through the food web [4]. The toxicity of gambieric acids C and D and their role in CP is not clear, the only toxicological data is in mouse lymphoma cells L5178Y showing a moderately low toxic effect [17,30].

Despite the detection of a putative gambieroxide in the MSF of all strains of *G. australes* by LC-HRMS, its confirmation is not conclusive due to the lack of authentic gambieroxide to compare retention time as well as fragmentation in targeted MS/MS. Gambieroxide was first isolated from the *G. toxicus* GTP2 strain collected at Papeete in French Polynesia and has a structure similar to yessotoxin [18].

The only *F. paulensis* analyzed in this work was also from Menorca (Balearic Islands, Spain) and showed a low amount of MTX3, 10.5 pg MTX eq./cell, which may explain the poor recovery of 18% from the MSF, reinforcing the inadequacy of this liquid/liquid partition for this particular compound. MTX3 was also detected by LC-MS/MS in *F. paulensis* from Australia and New Zealand [31,32]. In the Western Mediterranean Sea (Formentera, Balearic Islands, Spain), a positive response was observed in the mouse bioassay (MBA) for both MSF and DSF of a *F. paulensis* and the inconclusive detection by LC-HRMS of gambieric acid A and 54-deoxy-CTX1B in trace levels [25].

*Gambierdiscus* sp.2 from Crete (Greece) showed a profile with low MTX3 concentration (4.2 pg MTX1-eq./cell) and gambierone at much higher concentration (776 pg MTX1-eq./cell). The biological activity of gambierone is similar to CTX3C but of lower potency (similar to MTX3) and since approximately 8% partitioned into the DSF, it should also be monitored in fish tissue from CP cases [15]. The presence of *Gambierdiscus* in Crete Island was first reported in 2003 by [24], while it was not until 2017 that a putative MTX3 was detected by LC-MS/MS in a *G. carolinianus* strain from this region [20].

The availability of a strain of *G. excentricus* from La Gomera (Canary Islands, Spain) allowed for confirmation that MTX4 is the main MTX present in this species as previously reported [11]. The liquid/liquid partitioning step appeared adequate for MTX4 as 99.3% of MTX4 were recovered in the MSF.

## 4. Conclusions

This study allowed for the characterization of the toxins produced by different species of dinoflagellates from the Mediterranean Sea. LC-MS/MS was used for a first and rapid screening, allowing a quantitative estimation of the toxins involved, while LC-HRMS allowed for their confirmation and characterization. Dinoflagellates from the Mediterranean Sea showed toxin profiles similar to those detected in CP endemic regions. Strains of *G. australes* from Mallorca and Menorca (Spain, Mediterranean Sea) produced MTX3, putative gambieric acid C and D, a putative gambierone analogue and putative gambieroxide, while MTX1 was absent in contrast to the same species from temperate waters of the Pacific Ocean. *F. paulensis* from the same region only produced MTX3, and the strain of *Gambierdiscus* sp.2 from Crete (Greece) produced MTX3 and gambierone. Further research needs to be carried out in order to evaluate the possible presence of CTXs in fish from the Mediterranean Sea, considering that the presence of these dinoflagellates is associated with the production and accumulation of CTXs through the marine food web.

## 5. Materials and Methods

### 5.1. Reference Toxins and Chemicals

Maitotoxin-1 (MTX1) standard used for the LC-MS analysis was obtained from Wako Chemicals USA, Inc. (Richmond, VA, USA). MTX1 standard was dissolved in MeOH:H_2_O (1:1, v/v) being the stock solution 10 µg·mL^−1^. MTX4 qualitative laboratory reference material partially purified from *Gambierdiscus excentricus* was available from a previous study at the Phycotoxins Laboratory [11]. MTX3 and gambierone were identified as both compounds have identical retention times in our chromatography in both *G. australes* and *G. belizeanus*, the two species from which they have been originally isolated. MTX3 and gambierone qualitative laboratory reference material from *G. australes* and *G. belizeanus* was available at the Phycotoxins Laboratory [11]. HPLC-grade methanol and dichloromethane for extraction were purchased from Sigma Aldrich (Saint Quentin Fallavier, France). Milli-Q water was supplied by a Milli-Q integral 3 system (Millipore, Saint-Quentin-Yvelines, France). Water, acetonitrile, formic acid and ammonium formate used to prepare mobile phases were of LC-MS grade. All these chemicals were purchased from Sigma Aldrich (Saint Quentin Fallavier, France).

In order to carry out the toxicity evaluation by Neuro-2a, CTX1B was provided by Dr. Lewis, University of Queensland and was stored in absolute methanol at −20 °C. Neuroblastoma murine cells were purchased in ATCC LGC standards (USA). Fetal bovine serum (FBS), L-glutamine solution, ouabain, veratridine, phosphate buffered saline (PBS), penicillin, streptomycin, RPMI-1640 medium, sodium pyruvate, thiazolyl blue tetrazolium bromide (MTT) were purchased from Merck KGaA (Darmstadt, Germany). Dimethyl sulfoxide (DMSO) and absolute methanol were purchased from Honeywell (Fürth, Germany) and from Chemlab (Zedelgem, Belgium), respectively. The incubator was purchased from Binder, Germany. The microplate reader KC4 was purchased from BIO-TEK Instruments, Inc.

### 5.2. Gambierdiscus and Fukuyoa Strains

All nine dinoflagellate extracts analyzed in this work were obtained from the collection of strains obtained in the EuroCigua project. All strains were cultivated at the IRTA laboratory (Tarragona, Spain) and the detailed information about these strains is shown in Table 4.

#### Culturing, Harvesting, Toxin Extraction of Gambierdiscus and Fukuyoa, and N2a Assay

All strains were inoculated in 5 L of medium ES (Provasoli 1968, modified by Jorge Diogène) and salinity was adjusted to 36 in 8 L flat-bottom, round glass-flasks. Cultures were maintained in filtered air, and light turbulence (gentle bubbling) was supplied by an air-pump system. The initial concentration of dinoflagellates was between 25 and 50 cells/mL. Strains were incubated at 24 ± 0.5 °C. The illumination was provided by fluorescent tubes with photon irradiance of 100 µmol m^−2^ s^−1^ under a 12:12 h L:D photoperiod. When cultures arrived at the late-exponential phase (after 20 ± 3 days), cultures were vigorously shaken and 15 mL aliquots were taken and fixed with Lugol’s iodine solution (3%) to estimate the cell concentration (cell/mL). Subsequently, the remaining volume of each strain was filtered and collected through a 10 µm plankton net (Holmbladsvej, Denmark) in sterile 50 mL Falcon tubes and centrifuged at 4300 *g* for 20 min (Allegra X-15R, Beckman Coulter). Supernatants were discarded and micro-algal pellets of each strain were pooled in one 50 mL Falcon tube. Centrifugation was repeated and supernatants were discarded. Pellets were subsequently kept at –20 °C with absolute methanol (10 mL for 10^6^ cells) until toxin extraction. To extract the toxin from micro-algal pellets, each pellet in methanol was sonicated using an ultrasonic cell disrupter (Watt ultrasonic processor VCX750, USA). The tip amplitude was set at 37% 3 sec on/3 sec off for 15 min. The sample was then centrifuged at 600 g for 5 min at 4 °C. Supernatant was then transferred to a glass vial. The procedure was repeated twice, one with methanol and another with aqueous methanol (50:50; v:v) (10 mL for 10^6^cells) and these were pooled. After that, the pool was evaporated to dryness with a rotary evaporator (Büchi Syncore, Switzerland) or dried under N₂ gas (Turbovap, Caliper, Hopkinton, USA) at 40 °C and re-suspended in pure methanol. These extracts were filtered with PTFE filters (0.2 µm) and stored at −20 °C.

The Neuro-2a assay was performed according to Reverté et al., (2018) [23].

### 5.3. Sample Pretreatment

The methanol extract from the cell pellet extraction was evaporated to dryness under N_2_ stream at 50 °C and CTX- and MTX-like compounds were partitioned as previously described [11]. Briefly, the residue from extraction was reconstituted in dichloromethane (50 mL/1 million cells) and partitioned twice with MeOH:H_2_O (3:2, v/v) (25 mL/1 million cells). Both organic and aqueous layers were evaporated to dryness under N_2_ stream at 50 °C and kept at −20 °C prior to the analysis. MTX-like compounds were supposed to partition into the MeOH:H_2_O (3:2, v/v), whereas CTX-like compounds were supposed to partition into the dichloromethane layer. Dried residue from the aqueous methanol fraction was reconstituted in 0.5 mL of MeOH:H_2_O (1:1, v/v), whereas the solid residue from the dichloromethane layer was reconstituted in 0.5 mL MeOH, with both being filtrated through 0.22 µm prior to the LC-MS analysis.

### 5.4. LC-MS Analysis

#### 5.4.1. LC-MS/MS (API 4000 QTrap)

LC-MS/MS analysis to monitor specific MTX congeners and gambierone was performed using an LC system (UFLC XR Nexera, Shimadzu, Japan) coupled to a hybrid triple quadrupole/ion-trap mass spectrometer API 4000 QTrap (SCIEX, Redwood City, CA, USA) equipped with a turboV^®^ ESI source. Maitotoxins and gambierone were separated using a reversed-phase C18 Kinetex column (100 Å, 2.6 µm, 50 × 2.1 mm, Phenomenex, Le Pecq, France) with water (A) and 95% acetonitrile/water (B), both containing 2 mM of ammonium formate and 50 mM of formic acid. The column oven and the sample tray temperatures were set at 40 °C and 4 °C, respectively. The flow rate was set at 0.4 mL min^−1^ and the injection volume was set to 5 µL. Separation was achieved using the following mobile phase gradient: from 10% to 95% B in 10 min, keep at 95% B for 2 min, return to 10% B in 0.1 min and equilibration for 3.9 min prior the next injection. The instrument control, data processing and analysis were conducted using Analyst software 1.6.3 (Sciex, Redwood city, CA, USA). LC-MS/MS analyses were carried out in negative ion acquisition mode, monitoring the transitions shown in Appendix A in Multiple Reaction Monitoring (MRM) mode with a dwell time of 80 ms Retention time of the different compounds with reference material available are shown in Appendix A. Source conditions were curtain gas 25 psi, ionspray −4.5 kV, turbogas temperature of 500 °C, gas 1 and 2 set at 50 psi, and an entrance and declustering potential of −10 V and −210 V, respectively. Positive ion acquisition mode was also used in the analysis of MTX3 and gambierone. Source conditions were curtain gas 25 psi, ionspray 4.5 kV, turbogas temperature of 500 °C, gas 1 and 2 set at 50 psi, and an entrance and declustering potential of 10 V and 100 V, respectively.

The fragment ion monitored in negative ionization mode for all the MRM transition of the MTX-group of toxins was the hydrogenated sulfate anion *m/z* 96.9 [HOSO_3_]^−^ which was used as the confirmatory transition. Quantification of MTX1, MTX4, desulfo-MTX1 and didehydro-demethyl-desulfo-MTX1 was conducted using the MRM transition [M−2H]^2−^/[M−2H]^2−^ for MTX3 and gambierone [M-H]^−^/ [M-H]^−^ (Appendix A). Due to the lack of the appropriate standards for the quantitation of each compound, MTX3, MTX4 and gambierone were quantified against the MTX1 calibration curve, assuming equal molar response and applying the same LOD and LOQ calculated for MTX1.

The MTX1 standard calibration range for the LC-MS/MS analysis consisted of seven concentrations ranging from 0.2 to 10 µg mL^−1^ in MeOH:H_2_O (1:1, v/v). Limit of detection (LOD) and quantification (LOQ) were determined with the ordinary least-squares regression data method [34,35]. The LOD was calculated as three times the standard deviation of the y-intercepts over the slope of the calibration curve; the LOQ was calculated as 10 times the standard deviation of the y-intercepts over the slope of the calibration curve [34,35]. Therefore, LOD and LOQ for the MTX1 MRM transition [M−2H]^2−^/[M−2H]^2−^ were 0.32 and 0.97 µg mL^−1^, respectively.

#### 5.4.2. LC-HRMS and HRMS/MS (Q-Tof 6550 iFunnel)

LC-HRMS analyses were carried out using a UHPLC system 1290 Infinity II (Agilent Technologies, Santa Clara, CA, USA) coupled to a HRMS time of flight mass spectrometer Q-Tof 6550 iFunnel (Agilent Technologies, Santa Clara, CA, USA). Chromatographic separation was performed using a Kinetex C18 column (100 Å, 1.7 µm, 100 × 2.1 mm, Phenomenex, Le Pecq, France) at 40 °C with water (A) and 95% acetonitrile (B) both containing 2 mM ammonium formate and 50 mM formic acid. The flow rate was 0.4 mL min^−1^ and the injection volume was 5 µL. Gradient of mobile phase was carried out as follows: 5% B was kept for 1 min, then increased to 100% B over 11 min, kept at 100% B for 2 min and returned to the initial conditions in 0.5 min and then equilibrated the column for 4.5 min prior to the next injection.

Source conditions were set as follows: gas temperature, 160 °C; gas flow, 11 L/min; nebulizer, 45 psi; sheath gas temperature, 250 °C; sheath gas flow, 11 L/min; capillary voltage, 4500 V and nozzle voltage, 500 V. The instrument was calibrated, using the Agilent tuning mix, in negative and positive ionization mode before each analysis.

LC-HRMS analyses were carried out in full scan and targeted MS/MS mode in positive and negative ionization mode in separate runs. Full scan analysis operated at a mass resolution of 40,000 Full width at Half Maximum (FWHM) over a mass-to-charge ratio (*m/z*) ranging from 100 to 3200 with a scan rate of 1 spectra/s. Targeted MS/MS was performed in a Collision Induced Dissociation (CID) cell at 45,000 FWHM over the scan rage from m/z 50 to 1700 with a scan rate of 10 spectra/s and a scan rate of 3 spectra/s applying three different collision energies in order to have a good fragmentation pathway. Two reference masses *m/z* 121.0509 (purine) and *m/z* 922.0099 (hexakisphosphazine) were continuously monitored during the entire run. Data acquisition was controlled by MassHunter software (Agilent technologies, CA, USA). Raw data were processed with Agilent MassHunter Qualitative Analysis software (version B.07.00, service pack 1) using the Find by Formula (FbF) algorithm screening with a Personal Compound Database and Library (PDCL) created by Phycotoxins laboratory (IFREMER, France).

## Figures and Tables

**Figure 1 toxins-12-00305-f001:**
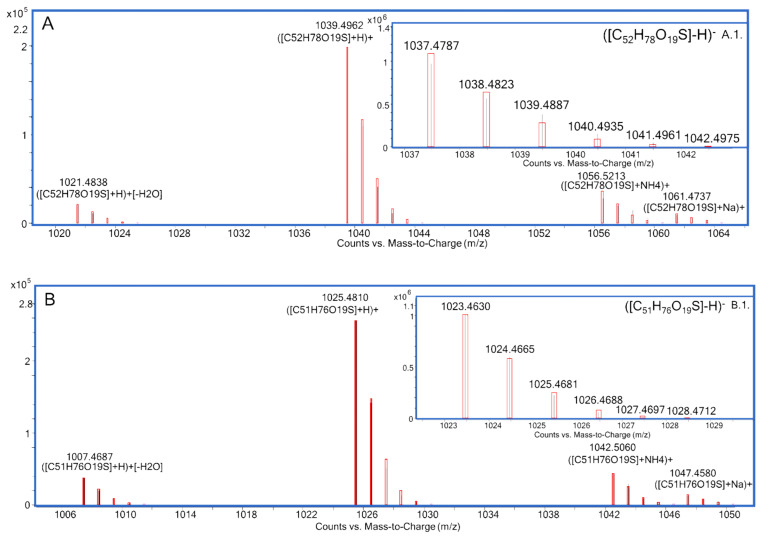
LC-HRMS analysis in MS full scan mode of: MTX3 detected in *G. australes*, (**A**) ESI^+^ mode, (**A1**) ESI^−^ mode; gambierone detected in *Gambierdiscus* sp.2, (**B**) ESI^+^ mode, (**B1**) ESI^−^ mode.

**Figure 2 toxins-12-00305-f002:**
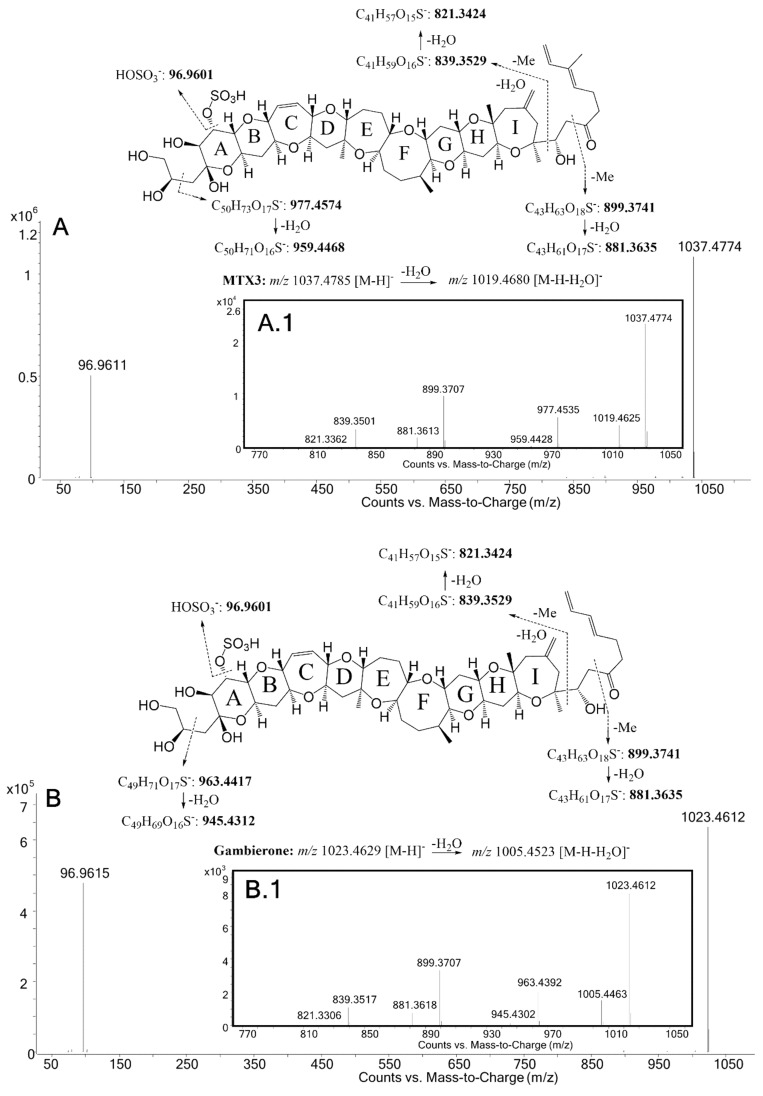
ESI-targeted HRMS/MS spectra of: MTX3 in *G. australes*: (**A**) average CE of 30 eV, 50 and 70 eV; (**A1**) zoom from *m/z* 770 to *m/z* 1050 at 70 eV, gambierone in *Gambierdiscus* sp.2; (**B**) average CE of 30 eV, 50 and 70 eV; (**B1**) zoom from *m/z* 770 to *m/z* 1050 at 70 eV.

**Figure 3 toxins-12-00305-f003:**
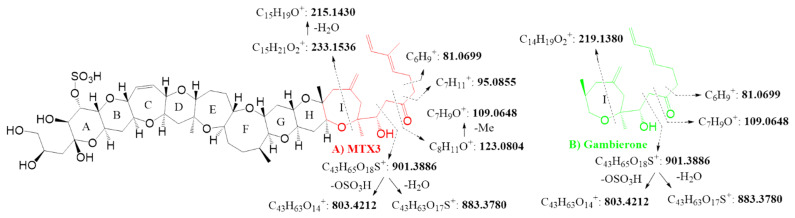
+ESI averaged 20 eV, 40 eV and 60 eV and targeted HRMS/MS spectra of: (**A**) MTX3 in *G. australes*, (**B**) gambierone in *Gambierdiscus.* sp.2.

**Figure 4 toxins-12-00305-f004:**
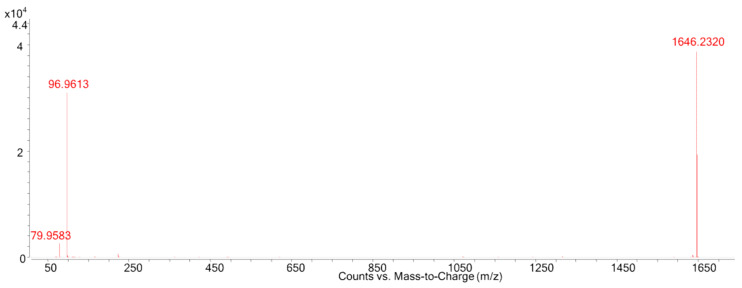
ESI^−^-targeted HRMS/MS spectra at 110 eV of MTX4 from *G. excentricus* extract.

**Figure 5 toxins-12-00305-f005:**
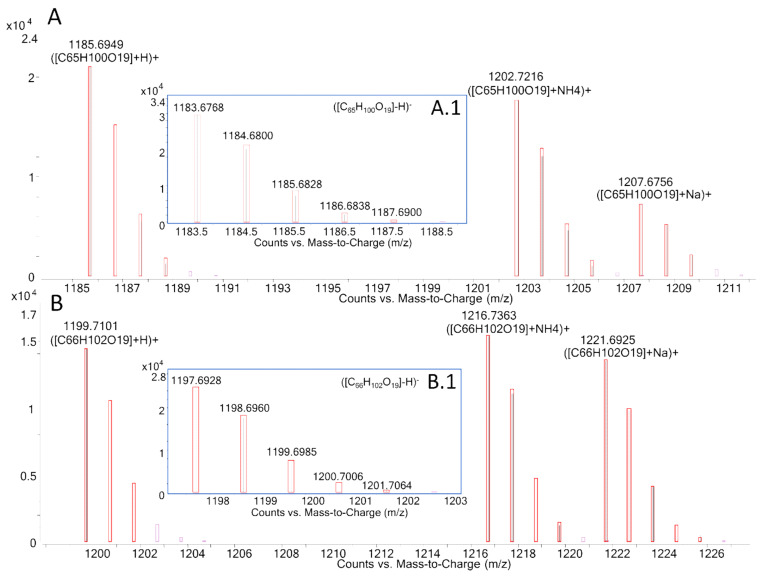
LC-HRMS full scan analysis of putative gambieric acid C, (**A**) ESI^+^, (**A1**) ESI^−^; putative gambieric acid D, (**B**) ESI^+^, (**B1**) ESI^−^ in *G. australes* extract.

**Figure 6 toxins-12-00305-f006:**
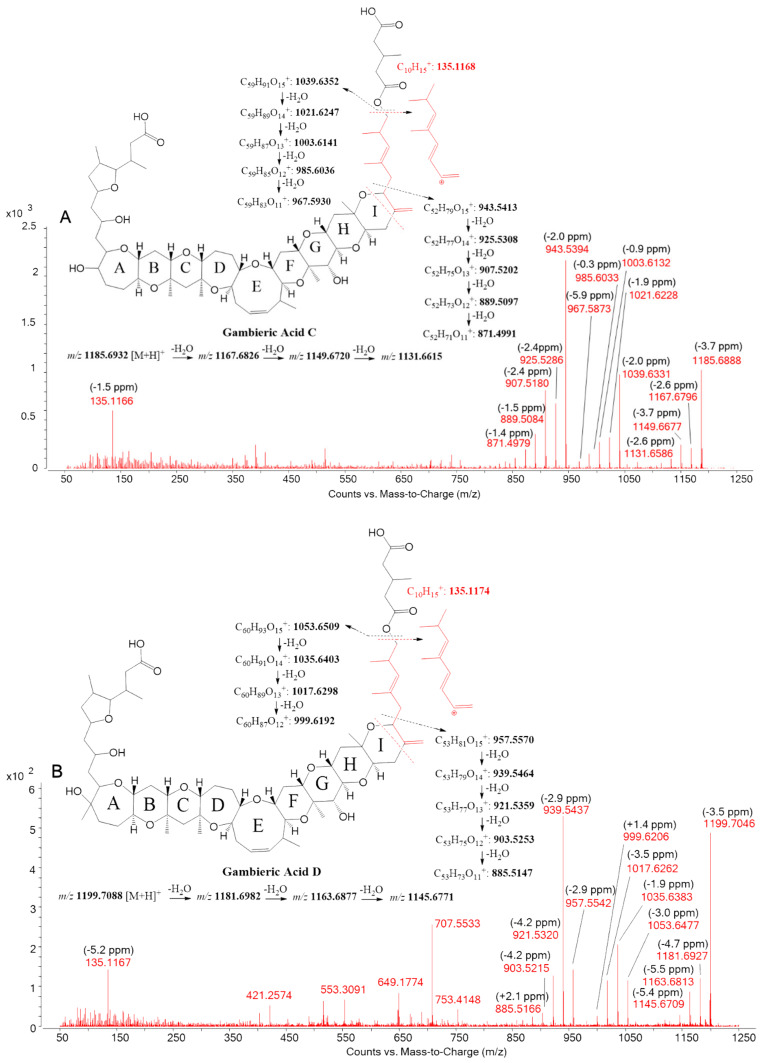
ESI^+^-targeted HRMS/MS spectra at an average of 20, 40 and 60 eV of (**A**) putative gambieric acid C and (**B**) putative gambieric acid D; both detected in *G. australes.*

**Table 1 toxins-12-00305-t001:** List of the maitotoxins isolated from dinoflagellates of the genus *Gambierdiscus* and *Fukuyoa.*

MTX Congener	Chemical Formula	Monoisotopic Mass (Da)	Reference
MTX1	C_164_H_258_O_68_S_2_	3379.6171 for the free acid form	[9]
MTX2	Unknown	3298 for the mono-sodium salt	[10]
MTX3	C_52_H_78_O_19_S	1038.4858 for the free acid form	[13,14]
MTX4	Unknown	3292.486 for the free acid form	[11]
desulfo-MTX1	C_164_H_258_O_65_S	3299.6603 for the free acid form	[12]
didehydro-demethyl-desulfo-MTX1	C_163_H_254_O_65_S	3283.6290 for the free acid form	[12]

**Table 2 toxins-12-00305-t002:** Results obtained after the liquid chromatography mass spectrometry (LC-MS/MS) analysis of the dinoflagellate extracts using a triple-stage quadrupole instrument. Results are expressed in pg MTX1 eq./cell: equivalent/cell; MSF: Methanol Soluble Fraction; DSF: Dichloromethane Soluble Fraction; Σ: sum, n.d.: not detected.

Species	Strain Code	Location					pg MTX1 eq./Cell (% of TOTAL)											
			MTX1	MTX2	MTX3			MTX4			desulfo-MTX1	didehydro-demethyl-desulfo-MTX1	Gambierone			P-Gambierone Analogue		
					MSF (%)	DSF (%)	Σ	MSF (%)	DSF (%)	Σ			MSF (%)	DSF (%)	Σ	MSF (%)	DSF (%)	Σ
*G. australes*	IRTA-SMM-17-189	Torret, Menorca, Balearic Islands, Spain	n.d.	n.d.	241 (70.2)	102 (29.8)	344.0	n.d.			n.d.	n.d.	n.d.			148 (99.7)	0.4 (0.3)	149
*G. australes*	IRTA-SMM-17-162	St. Adeodat, Menorca, Balearic Islands, Spain	n.d.	n.d.	575 (79.9)	150 (20.1)	720	n.d.			n.d.	n.d.	n.d.			275 (99.6)	1.0 (0.4)	276
*G. australes*	IRTA-SMM-17-164	St. Adeodat, Menorca, Balearic Islands, Spain	n.d.	n.d.	1108 (66.7)	553 (33.3)	1661	n.d.			n.d.	n.d.	n.d.			522 (99.8)	1.2 (0.2)	523
*G. australes*	IRTA-SMM-17-271	Macarella, Menorca, Balearic Islands, Spain	n.d.	n.d.	1107 (83.7)	215 (16.3)	1322	n.d.			n.d.	n.d.	n.d.			408 (99.9)	0.4 (0.1)	409
*F. paulensis*	IRTA-SMM-17-209	Sacaleta, Menorca, Balearic Islands, Spain	n.d.	n.d.	1.8 (17.7)	8.6 (82.3)	10.5	n.d.			n.d.	n.d.	n.d.			n.d.		
*G. australes*	IRTA-SMM-17-253	Anguila, Mallorca, Balearic Islands, Spain	n.d.	n.d.	781 (72.3)	300 (27.7)	1081	n.d.			n.d.	n.d.	n.d.			229 (99.8)	0.5 (0.2)	229
*G. australes*	IRTA-SMM-17-244	Camp de Mar, Mallorca, Balearic Islands, Spain	n.d.	n.d.	403 (84.1)	75.9 (15.9)	479	n.d.			n.d.	n.d.	n.d.			173 (99.6)	0.7 (0.4)	174
*Gambierdiscus* sp*.2*	0010G-CR-CCAUTH	Kolimpari, Crete, Greece	n.d.	n.d.	3.0 (70.1)	1.3 (29.9)	4.3	n.d.			n.d.	n.d.	716 (92.2)	60.3 (7.8)	775.9	n.d.		
*G. excentricus*	IRTA-SMM-17-407	Playa de vueltas, La Gomera, Canary Islands, Spain	n.d.	n.d.	n.d.			36.8 (99.3)	0.3 (0.7)	37.1	n.d.	n.d.	n.d.			n.d.		

**Table 3 toxins-12-00305-t003:** Accurate masses (measured and theoretical) of informative ions of MTX3 and gambierone in ESI^−^ mode and ESI^+^ mode.

Ion	MTX3				Gambierone			
	Molecular Formula	m/z		Δ ppm	Molecular Formula	m/z		Δ ppm
		Measured	Theoretical			Measured	Theoretical	
**ESI^−^**								
**[M-H]^−^**	**C_52_H_77_O_19_S^−^**	**1037.4774**	1037.4785	−1.1	C_51_H_75_O_19_S^−^	1023.4612	1023.4629	−1.6
[M-H-H_2_O]^−^	C_52_H_75_O_18_S^−^	1019.4625	1019.4680	−5.3	C_51_H_73_O_18_S^−^	1005.4463	1005.4523	−6.0
	C_50_H_73_O_17_S^−^	977.4535	977.4574	−4.0	C_49_H_71_O_17_S^−^	963.4392	963.4417	−2.6
H_2_O	C_50_H_71_O_16_S^−^	959.4428	959.4468	−4.2	C_49_H_69_O_16_S^−^	945.4302	945.4312	−1.0
	C_43_H_63_O_18_S^−^	899.3707	899.3741	−3.7	C_43_H_63_O_18_S^−^	899.3707	899.3741	−3.7
-H_2_O	C_43_H_61_O_17_S^−^	881.3613	881.3635	−2.5	C_43_H_61_O_17_S^−^	881.3618	881.3635	−1.9
	C_41_H_59_O_16_S^−^	839.3501	839.3529	−3.4	C_41_H_59_O_16_S^−^	839.3517	839.3529	−1.5
-H_2_O	C_41_H_57_O_15_S^−^	821.3362	821.3424	−7.5	C_41_H_57_O_15_S^−^	821.3306	821.3424	−14.3
[HOSO_3_]^−^	HOSO_3_^−^	96.9611	96.9601	10.3	HOSO_3_^−^	96.9615	96.9601	14.4
**ESI^+^**								
[M+H]^+^	C_52_H_79_O_19_S^+^	1039.4911	1039.4931	−1.9	C_51_H_77_O_19_S^+^	1025.4749	1025.4774	−2.4
[M+H-H_2_O]^+^	C_52_H_77_O_18_S^+^	1021.4795	1021.4825	−2.9	C_51_H_75_O_18_S^+^	1007.4636	1007.4668	−3.2
[M+H-2H_2_O]^+^	C_52_H_75_O_17_S^+^	1003.4692	1003.4720	−2.8	C_51_H_73_O_17_S^+^	989.4528	989.4563	−3.5
[M-SO_3_+H]^+^	C_52_H_79_O_16_^+^	959.5333	959.5363	−3.1	C_51_H_77_O_16_^+^	945.5171	945.5206	−3.7
[M-SO_3_-H_2_O+H]^+^	C_52_H_77_O_15_^+^	941.5232	941.5257	−2.7	C_51_H_75_O_15_^+^	927.5070	927.5100	−3.2
[M-SO_3_-2H_2_O+H]^+^	C_52_H_75_O_14_^+^	923.5125	923.5152	−2.9	C_51_H_73_O_14_^+^	909.4968	909.4995	−3.0
[M-SO_3_-3H_2_O+H]^+^	C_52_H_73_O_13_^+^	905.5022	905.5046	−2.7	C_51_H_71_O_13_^+^	891.4849	891.4889	−4.5
[M-SO_3_-4H_2_O+H]^+^	C_52_H_71_O_12_^+^	887.4911	887.4940	−3.3	C_51_H_69_O_12_^+^	873.4743	873.4783	−4.6
[M-SO_3_-5H_2_O+H]^+^	C_52_H_69_O_11_^+^	869.4812	869.4835	−2.6	C_51_H_67_O_11_^+^	855.4656	855.4678	−2.6
	C_43_H_65_O_18_S^+^	901.3854	901.3886	−3.6	C_43_H_65_O_18_S^+^	901.3884	901.3886	−0.2
-H_2_O	C_43_H_63_O_17_S^+^	883.3756	883.3780	−2.7	C_43_H_63_O_17_S^+^	883.3744	883.3780	−4.1
-OSO_3_H	C_43_H_63_O_14_^+^	803.4182	803.4212	−3.7	C_43_H_63_O_14_^+^	803.4173	803.4212	−4.9
	C_15_H_21_O_2_^+^	233.1533	233.1536	−1.3	C_14_H_19_O_2_^+^	219.1375	219.1380	−2.1
	C_15_H_19_O^+^	215.1429	215.1430	−0.5	C_7_H_9_O^+^	109.0644	109.0648	−3.6
	C_8_H_11_O^+^	123.0800	123.0804	−3.2	C_6_H_9_^+^	81.0697	81.0699	−2.2
	C_7_H_9_O^+^	109.0645	109.0648	−2.8				
	C_7_H_11_^+^	95.0852	95.0855	−3.2				
	C_6_H_9_^+^	81.0696	81.0699	−3.7				

**Table 4 toxins-12-00305-t004:** Detailed information about the dinoflagellate extracts analyzed with the neuroblastoma cell-based assay.

Species	Strain Code	Location	Number of Cells Extracted	Volume of Culture (L)	CTX-like (fg CTX1B Equiv./Cell)
*G. australes*	IRTA-SMM-17-189	Torret, Menorca, Balearic Islands, Spain	17 134 000	20	83 ± 12 ^a^
*G. australes*	IRTA-SMM-17-162	St. Adeodat, Menorca, Balearic Islands, Spain	27 811 000	20	101 ± 7.5
*G. australes*	IRTA-SMM-17-164	St. Adeodat, Menorca, Balearic Islands, Spain	4 257 000	20	>62.5 (NQ)
*G. australes*	IRTA-SMM-17-271	Macarella, Menorca, Balearic Islands, Spain	14 007 000	20	271 ± 29
*F. paulensis*	IRTA-SMM-17-209	Sacaleta, Menorca, Balearic Islands, Spain	6 964 000	20	16 ± 1.7 ^a^
*G. australes*	IRTA-SMM-17-253	Anguila, Menorca, Balearic Islands, Spain	13 735 000	20	164 ± 16
*G. australes*	IRTA-SMM-17-244	Camp de Mar, Mallorca, Balearic Islands, Spain	4 121 000	5	155 ± 25
*Gambierdiscus* sp.2	0010G-CR-CCAUTH	Kolimpari, Crete, Greece	2 300 000	5	NQ
*G. excentricus*	IRTA-SMM-17-407	Playa de vueltas, La Gomera, Canary Islands, Spain	6 084 000	5	>794 (NQ)

NQ: not quantifiable; ^a^ CTX-like toxicity evaluated in [33].

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
