# Peer review of "Use of Mass Spectrometry to Determine the Diversity of Toxins Produced by Gambierdiscus and Fukuyoa Species from Balearic Islands and Crete (Mediterranean Sea) and the Canary Islands (Northeast Atlantic)"

_toxins, 2020, doi:10.3390/toxins12050305_

Round 1

Reviewer 1 Report

Authors investigated the contents of MTX analogs from various Gambierdiscus spp. and Fukuyoa spp. obtained from European countries using LC-MS techniques. The results are informative to  researchers dealing with MTXs and the manuscript should be accepted to the journal. However, a problem on determination of toxins should be clarified before the acceptance.

 Authors determined some compounds from only m/z numbers of compounds observed in the LC-MS data but not from retention times. There are no reference standards for MTX2, 3, desulfo-MTX1 and didehydro-demethyl-desulfo-MTX1 as their retention times would not be obtained. Thus, the determination of these compound are impossible. For MTX3, authors discussed  the MS/MS fragment of the compound without considering the alternative structure, ie, structural isomer at the side chain. In addition, reference standard of MTX4 is not reliable since its structure has not been elucidated.

Thus, the identification of MTX analogs, except for MTX1, performed in this study should be recognized as a speculation. Authors should classify their results into facts and speculations and re-organize the manuscript. From above reasons, the data of gambieric acids C and D should not be discussed without comparison with its reference standard.

Author Response

Authors investigated the contents of MTX analogs from various Gambierdiscus spp. and Fukuyoa spp. obtained from European countries using LC-MS techniques. The results are informative to researchers dealing with MTXs and the manuscript should be accepted to the journal. However, a problem on determination of toxins should be clarified before the acceptance.

The authors thank the reviewer for the overall appreciation and reply to specific comments below.

Authors determined some compounds from only m/z numbers of compounds observed in the LC-MS data but not from retention times. There are no reference standards for MTX2, 3, desulfo-MTX1 and didehydro-demethyl-desulfo-MTX1 as their retention times would not be obtained. Thus, the determination of these compound are impossible. For MTX3, authors discussed the MS/MS fragment of the compound without considering the alternative structure, ie, structural isomer at the side chain. In addition, reference standard of MTX4 is not reliable since its structure has not been elucidated.

We agree with the reviewer that the lack of reference compounds in this field of research hampers progress. We have, however, undertaken a number of steps to ensure a maximum of confidence in our results:

  • We agree that there is too little information for proper identification of MTX2. However, we only report the absence of this compound as we could not identify any masses corresponding to this compound. Proof of absence is related to limit of detection which we could not assess due to the lack of reference standards being available. We stated this more explicitly in the manuscript. Lines 122-130 (of the revised manuscript in simple markings-mode): “On the other hand, the lack of structural information and reference material of MTX2 limited its identification. Therefore, the absence of this compound was reported as the not detection above the limit of detection of any ion transition monitored.”
  • For desulfo-MTX1 and didehydro-demethyl-desulfo-MTX1, accurate masses are available and therefore, their absence in or samples is only hampered by our inability to derive appropriate limits of detection. We stated this more explicitly in the manuscript. Lines 122-130 (of the revised manuscript in simple markings-mode): “Also, the lack of reference material of desulfo-MTX1 and didehydro-demethyl-desulfo-MTX1 limited its identification being the approach followed the same as the described for MTX2. However, the accurate mass of these compounds are available only hampering its identification in LC-HRMS by the uncertainty of the limit of detection of our method for these compounds.”
  • MTX3, alias 44-methyl gambierone has been reported independently by two groups (Boente-Juncal et al., 2019, and Murray et al., 2019), both of whom we already cited in the manuscript as references [13] and [14], respectively. This compound has been fully identified after preparative isolation from two separate species of Gambierdiscus ( belizeanus and G. australes), and both these species were available to us in our laboratory. Therefore, it is reasonable to assume that the compound which has the correct accurate mass and the same retention time in both species is authentic MTX3. Therefore, we have clear evidence toward the compound we assume to be MTX3 actually being MTX3. We were not explicit enough about our approach in identifying MTX3 as such and have amended this accordingly in the manuscript: Lines 122-123 (of the revised manuscript in simple markings-mode): “The identification of MTX1, MTX3, gambierone and MTX4 was carried out by comparing retention times as well as ion ratios with the reference materials available (Table S8). Also in M&M (section 5.4.1, lines 464-465 (of the revised manuscript in simple markings mode): “Retention time of the different compounds with reference material available are shown in Table S8”
  • We acknowledge that the structure of MTX4 has not been fully elucidated. Still, since we have this reference material we were able to compare accurate mass and retention time, irrespective of the structure of the compound. We didn’t feel the manuscript requires further clarification than what was given in lines 161-164 (in the revised manuscript with simple markings-mode):” MTX4 was confirmed in a strain of excentricus from La Gomera (Canary Islands, Spain). Retention time as well as MRM ion ratios transitions m/z 1646.2 [M−2H]2−/m/z 1646.2 [M−2H]2− and m/z 1646.2 [M−2H]2−/m/z 96.9 [HOSO3] were consistent with those obtained in the MTX4 reference material.”. We also added in lines 390-391 (of the revised manuscript in simple markings-mode): “MTX4 qualitative laboratory reference material partially purified from Gambierdisus excentricus was available from a previous study at the Phycotoxins Laboratory [11].“

Thus, the identification of MTX analogs, except for MTX1, performed in this study should be recognized as a speculation. Authors should classify their results into facts and speculations and re-organize the manuscript. From above reasons, the data of gambieric acids C and D should not be discussed without comparison with its reference standard.

We stated more clearly the limitations of our study regards MTX2, desulfo-MTX1 and didehydro-demethyl-desulfo-MTX1, and modified the text describing our approach in identifying MTX3 and MTX4 (lines 122-130 of the revised manuscript in simple markings mode). Text of the manuscript has been revised throughout regarding putative gambieric acids C and D and the text now includes that our study is a merely a tentative identification of these compounds (abstract and conclusions).

Reviewer 2 Report

The manuscript describes the toxin LCMS/MS profiles of dinoflagellates of two genera from three different locations. Even though the LCMS/MS analyses were done well and well described, the main goal in the manuscript is hard to understand and the article requires additional inputs. In my opinion, in the current state, the article is not suitable for publication.

Please find my comments below:

1) First of all, the title is way too long.
- The authors do not need to include the analytical tools used to describe the chemistry of dinoflagellates as we do expect the use of such techniques for this kind of study
- The authors include "characterize the diversity of metabolites" in the title while only the toxin content has been described in this work 

2) The study is not well elaborated, leading to a manuscript hard to understand. The use of both low and high resolution mass spectrometry for this kind of work is not appropriate as the main objective of the article is to characterize the diversity of compounds and therefore only the high res should be used for such things. 

3) A major part of the discussion is based on the toxin content comparison of the three locations with CP occuring areas. A metabolomic study would have been perfectly feasable and more appropriate for this work and of greater value than quantification.

4) There are some issues with calibration of the MS for the putative gambierone analysis. Such delta variation between the mass of gambierone and the putative analogue is soly due to a wrong calibration.

5) A deeper analysis and desciption of the fragmentation pattern of both putative analogues (gambierone and gambieroxide) should be performed. The authors could propose a structure based on these precious data.

Author Response

The manuscript describes the toxin LCMS/MS profiles of dinoflagellates of two genera from three different locations. Even though the LCMS/MS analyses were done well and well described, the main goal in the manuscript is hard to understand and the article requires additional inputs. In my opinion, in the current state, the article is not suitable for publication.

The authors thank the reviewer for his appreciation of our LC-MS/MS study and reply to individual comments below.

Please find my comments below:

1) First of all, the title is way too long.

The author instructions of the journal Toxins do not specify a maximum length of title (while they do specify a maximum length of the abstract in number of words):

Title: The title of your manuscript should be concise, specific and relevant. It should identify if the study reports (human or animal) trial data, or is a systematic review, meta-analysis or replication study. When gene or protein names are included, the abbreviated name rather than full name should be used.  

Therefore, we believe the appropriateness of the length of the title is subjective. Even so, we agree that our concise title was unusually long.

- The authors do not need to include the analytical tools used to describe the chemistry of dinoflagellates as we do expect the use of such techniques for this kind of study

Leaving out the techniques used in the study would mislead the reader to understand that we may have isolated the metabolites from these strains and we therefore prefer keeping the reference to mass spectrometry.

- The authors include "characterize the diversity of metabolites" in the title while only the toxin content has been described in this work 

Taking on board this last comment, we would like to propose a shorter title as follows:

Use of mass spectrometry to determine the diversity of toxins produced by Gambierdiscus and Fukuyoa strains from the Balearic Islands, Crete (Mediterranean Sea) and the Canary Islands (Northeast Atlantic)

2) The study is not well elaborated, leading to a manuscript hard to understand. The use of both low and high resolution mass spectrometry for this kind of work is not appropriate as the main objective of the article is to characterize the diversity of compounds and therefore only the high res should be used for such things. 

The comment is not very precise. While we agree that there is higher value in the use of high resolution mass spectrometry in the identification of compounds for which no reference compounds are available we are also aware that high resolution mass spectrometry is not available to many laboratories. Therefore, we also believe that there is scientific merit in using and presenting both approaches as this allows to both clarify the limitations of low resolution mass spectrometry and make the study more comparable and repeatable to other laboratories.

Therefore, we re-wrote the end of the introduction to read: “The aim of the present study was to describe the diversity of toxins produced by a selection of  strains of Gambierdiscus and Fukuyoa species using liquid chromatography coupled to low and high resolution mass spectrometry (LC-MS/MS) and (LC-HRMS). While LC-MS/MS was used for a rapid screening of the toxic compounds potentially present, also allowing the estimation of their concentrations, the purpose of LC-HRMS was the confirmation and structural characterization of the toxic compounds previously identified by LC-MS/MS.”.

Furthermore, in the conclusion section of the manuscript we reformulated the first sentence to read: “This study allowed the characterization of the toxins produced by different species of dinoflagellates from the Mediterranean Sea. LC-MS/MS was used for a first and rapid screening also allowing a quantitative estimation of the toxins involved, while LC-HRMS allowed their confirmation and characterization”.

3) A major part of the discussion is based on the toxin content comparison of the three locations with CP occuring areas. A metabolomic study would have been perfectly feasable and more appropriate for this work and of greater value than quantification.

We agree that the initial title was somewhat misleading since we mentioned “metabolites” rather than “toxins” of Gambierdiscus & Fukuyoa strains. We have changed the title accordingly and we feel that the LC-MS/MS and HRMS study we presented is appropriate for the conclusions we draw. Suggestions for further studies could be numerous and we did not want to be too speculative/suggestive.

4) There are some issues with calibration of the MS for the putative gambierone analysis. Such delta variation between the mass of gambierone and the putative analogue is soly due to a wrong calibration.

We do not appreciate the suggestion of “wrong calibration”. The instrument was calibrated before each batch of analysis and in the Agilent Technology, reference compounds are infused continuously to ensure stability of this calibration over the run as mentioned in the M&M section of our manuscript (lines 506 to 508 of the revised manuscript in simple markings mode).

We introduced the analogue of gambierone in the results section where we describe the use of the low resolution tandem mass spectrometry (line 145 and following of the revised manuscript in simple markings mode). We know from your comment above that you had initially not appreciated our use of the low resolution tandem MS but we maintain that this is an important part of our study as it makes it comparable to the difficulties other laboratories face that do not have access to HRMS. Figure S1 clearly shows that a rapid assessment in low resolution mass spectrometry would have easily misidentified this analog for gambierone while the Table S4 presenting HRMS data clearly shows that the compound is significantly different from gambierone, irrespective of the slightly larger variations as expected for gambierone.

5) A deeper analysis and desciption of the fragmentation pattern of both putative analogues (gambierone and gambieroxide) should be performed. The authors could propose a structure based on these precious data.

We do not believe it is appropriate to suggest structures from HRMS spectra alone.

Round 2

Reviewer 1 Report

The compound determination by LC-HRMS and LC-MS/MS without reference standard would not be recognized as "determination" or "identification" of the compound since the existence of compound with alternative structure can be occur. Determination of compound in analytical method should introduce reference standard.

If the results should be published quickly in order to save public health, it should be stated in the manuscript and accepted to the journal. However, the results are not likely need to be published urgently. Authors should classify their results into the facts and the speculations and re-organize the manuscript.